# Fecal Microbiome Differences in Angus Steers with Differing Feed Efficiencies during the Feedlot-Finishing Phase

**DOI:** 10.3390/microorganisms10061128

**Published:** 2022-05-31

**Authors:** Jeferson M. Lourenco, Christina B. Welch, Taylor R. Krause, Michael A. Wieczorek, Francis L. Fluharty, Michael J. Rothrock, T. Dean Pringle, Todd R. Callaway

**Affiliations:** 1Department of Animal and Dairy Science, University of Georgia, Athens, GA 30602, USA; christina.welch@uga.edu (C.B.W.); taylor.krause@uga.edu (T.R.K.); michael.wieczorek@uga.edu (M.A.W.); ffluharty@uga.edu (F.L.F.); dpringle@uga.edu (T.D.P.); todd.callaway@uga.edu (T.R.C.); 2Egg Safety and Quality Research Unit, Richard B. Russell Research Center, Agricultural Research Service, USDA, Athens, GA 30605, USA; michael.rothrock@usda.gov

**Keywords:** feed efficiency, feedlot-finishing phase, fecal microbiota, Angus steers

## Abstract

The gastrointestinal microbiota of cattle is important for feedstuff degradation and feed efficiency determination. This study evaluated the fecal microbiome of Angus steers with distinct feed efficiencies during the feedlot-finishing phase. Angus steers (*n* = 65), fed a feedlot-finishing diet for 82 days, had growth performance metrics evaluated. Steers were ranked based upon residual feed intake (RFI), and the 5 lowest RFI (most efficient) and 5 highest RFI (least efficient) steers were selected for evaluation. Fecal samples were collected on 0-d and 82-d of the finishing period and microbial DNA was extracted and evaluated by 16S rRNA gene sequencing. During the feedlot trial, inefficient steers had decreased (*p* = 0.02) *Ruminococcaceae* populations and increased (*p* = 0.01) *Clostridiaceae* populations. Conversely, efficient steers had increased *Peptostreptococcaceae* (*p* = 0.03) and *Turicibacteraceae* (*p* = 0.01), and a trend for decreased *Proteobacteria* abundance (*p* = 0.096). Efficient steers had increased microbial richness and diversity during the feedlot period, which likely resulted in increased fiber-degrading enzymes in their hindgut, allowing them to extract more energy from the feed. Results suggest that cattle with better feed efficiency have greater diversity of hindgut microorganisms, resulting in more enzymes available for digestion, and improving energy harvest in the gut of efficient cattle.

## 1. Introduction

The gastrointestinal (GIT) microbiome of cattle helps them to digest feedstuffs by producing a myriad of enzymes that would otherwise not be present in the animal’s digestive process [1]. Recent studies have indicated that the GIT microbial population may be linked to improved animal performance [2,3,4,5]. Consequently, characterizing the microbial composition of the GIT of cattle and establishing its relationship with relevant animal performance metrics may allow us to use such microbial information in the prediction of those desirable traits. In addition, a better knowledge of the GIT microbiome (and how it impacts the host animal physiology) can provide direction in the development of novel feed additives, including potential probiotics.

In beef cattle operations, profit margins are small and feeding animals constitutes a major component of the total cost of production [6]. One way to increase profit margins is by improving feed efficiency, either through selective breeding or dietary manipulation. Evaluation of cattle efficiency can be based on residual feed intake (RFI). In simple terms, RFI determines whether an animal eats more or less feed than predicted for a given level of body weight and animal performance [7,8,9]. Therefore, animals with lower RFI values are more efficient in their use of feed than those with higher RFI.

In spite of the numerous potential benefits associated with a better understanding of cattle’s GIT microbiome, our knowledge remains limited [10]. Although the ruminal environment has been more extensively studied both in adult [3,4,11,12,13] and young cattle [14,15], fewer studies have focused on the hindgut microbial environment. However, there is evidence that the hindgut microbiome can also impact cattle performance and efficiency [16,17,18,19]. Therefore, the present study was designed to investigate the changes in the intestinal microbiome of Angus beef steers during the feedlot-finishing phase. Specifically, we investigated associations between these changes and different feedlot feed efficiencies, which would indicate that their intestinal microbial populations as a whole, or individual families, impact the efficiency by which cattle convert feed into body weight (muscle, bone, and fat).

## 2. Materials and Methods

### 2.1. Experimental Animals

All procedures involving animals were verified and approved by the University of Georgia’s Office of Animal Care and Use (AUP #A2012 11-006-R1). The present study was conducted as a smaller portion of a more expansive study [19]. Briefly, for five generations, commercial Angus cows were bred to high and low efficiency bulls. The goal was to determine the effect of selection using residual average daily gain (RADG) and marbling expected progeny differences (EPDs) on productivity, performance, and carcass quality from this selected herd. The current study utilized the fifth generation of steers born to that selection program and analyzed their performance in a commercial grain-fed feedlot-finishing system. The steers (*n* = 65) were fed in a commercial feedlot (Ridgefield Farm L.L.C.; Brasstown, NC, USA; 35.0391° N, 83.9576° W) from 8 March to 29 May 2018. Steers were rank ordered based upon their RFI data collected in the feedlot period, and then the 5 most efficient and 5 least efficient steers were selected for the present analysis of their fecal microbiomes. More information regarding their performance can be found in Appendix A.

Steers were adapted to the high grain feedlot finishing ration for 21-d prior to the trial, and all rations were formulated to meet the nutrient requirements for finishing cattle [20]. Over the 82-d feeding trial, daily feed intake was recorded using a GrowSafe feed intake monitoring system (GrowSafe Systems^®^ Ltd., Calgary, AB, Canada), which monitored individual feed intake. Steers were weighed at the beginning, mid-point, and end of the feedlot experimental period (d 0, 41, and 82, respectively). Using the individual dry matter intakes (DMI) and body weight gains for each steer, RFI was calculated as the difference between actual and expected DMI. The expected DMI was the residual estimate for each steer, following linear regression of midpoint, metabolic body weight, and average daily gain (ADG) on actual DMI.

### 2.2. Sample Collection, DNA Extraction, and Sequencing

Fecal samples were obtained aseptically via fecal grab from the rectum using separate palpation sleeves from all steers in the study twice: at the beginning (d 0) and end (d 82) of the feedlot-finishing phase. Upon collection, samples were immediately placed on ice, transported to the laboratory, and stored at −20 °C until further processing. Genomic DNA was extracted from the fecal samples following a semi-automated extraction procedure described by Rothrock Jr et al. [21]. Briefly, 0.33 g of sample was placed into a 2-mL Lysing Matrix E tube (MP Biomedicals LLC, Irvine, CA, USA), and both mechanical and enzymatic techniques were used to extract DNA, starting with the homogenization and disruption of cells via a FastPrep 24 Instrument (MP Biomedicals LLC, Irvine, CA, USA). Next, InhibitEX Tablets (QIAGEN, Venlo, The Netherlands) were added for enzymatic inhibition. The elution and purification of the DNA was completed using an automated robotic workstation (QIAcube; QIAGEN, Venlo, The Netherlands). The purity and concentration of the extracted DNA was evaluated using a Synergy H4 Hybrid Multi-Mode Microplate Reader along with the Take3 Micro-Volume Plate (BioTek Instruments Inc; Winooski, VT, USA). Samples were sorted based on meeting both of the following requirements: a minimum volume of 20 μL, and minimum concentration of 10 ng/μL of DNA. Samples that met these requirements were stored at 4 °C until further processing. Samples that failed to meet the requirements had to undergo a new DNA extraction cycle.

The extracted DNA samples were sent to the Georgia Genomics and Bioinformatics Core for 16S rRNA gene sequencing. Library preparation included PCR replications using the forward: S-D-Bact-0341-b-S-17 (5′-CCTACGGGNGGCWGCAG-3′); and reverse: S-D-Bact-0785-a-A-21 (5′-GACTACHVGGGTATCTAATCC-3′) primer pair [22]. The amplified DNA was then sequenced using an Illumina MiSeq instrument (Illumina Inc., San Diego, CA, USA).

### 2.3. 16S rRNA Gene Sequencing Data Analysis

The DNA sequence data was analyzed following the procedures described in Lourenco et al. [23]. Briefly, sequences were demultiplexed according to their barcodes, and converted into FASTQ files. Next, the high-quality pair-end reads were merged using BBMerge Paired Read Merger v37.64. Data files were then analyzed using the QIIME pipeline v1.9.1 [24]. Once cleared and merged, the files were converted to the FASTA format, and the sequences were clustered into operational taxonomic units (OTU) according to the Greengenes database (gg_13_8_otus). The OTU were defined at the threshold of 97% similarity. Following that, singleton OTU were removed, and the sequencing depth was set at 22,426 sequences per sample for further analyses. The following alpha-diversity indices were computed: Chao1 Index (microbial richness estimator) and Faith’s Phylogenetic Diversity Index (microbial diversity estimator). Additionally, relative bacterial abundances were quantified at the phylum and family levels.

### 2.4. Statistical Analysis

Analysis of the data was performed using Minitab v19.2 (Minitab LLC, State College, PA, USA). The animal performance data was analyzed by one-way ANOVA, using the animal efficiency status (i.e., more efficient or less efficient) as a fixed factor. The alpha-diversity indices and relative bacterial abundances were analyzed by paired *t*-tests, in which the periods (beginning or end of the feedlot phase) were compared according to the model:$$\;t=\frac{\bar{d}-\mu_{d_{0}}}{s_{\bar{d}}}$$
where $\bar{d}$ is the sample mean difference, $\mu_{d_{0}}$ is the hypothesized population mean difference, $s_{\bar{d}}$ = Sd/√n, *n* is the number of samples differences, and Sd is the standard deviation of the sample differences [25]. All results were considered significant when *p* ≤ 0.05 and were treated as trends when 0.05 < *p* ≤ 0.10.

## 3. Results

### 3.1. Animal Performance

Performance of the steers during the feedlot-finishing phase is summarized in Table 1. There was no difference in average daily gain (*p* = 0.82) between the most and the least efficient steers. Similarly, there was no difference regarding their final carcass weights (*p* = 0.62). Daily dry matter intake was significantly lower (*p* = 0.03) in the most efficient steers, compared to the less efficient steers. The metric used to evaluate their feed efficiency (RFI) revealed a significantly smaller value (*p* = 0.003) for the more efficient steers.

### 3.2. Alpha Diversity

Microbial richness (expressed as Chao1 index), and microbial diversity (expressed as Faith’s phylogenetic diversity index) are shown in Figure 1. Chao1 index significantly increased during the course of the feedlot in the most efficient steers (*p* = 0.04), but not in the less efficient steers (*p* = 0.28). Likewise, Faith’s phylogenetic diversity index was increased in the feces of the more efficient steers (*p* = 0.03), but not in the less efficient ones (*p* = 0.95).

### 3.3. Microbial Taxa

Major bacterial taxa (at phylum, family, and genus level) with changed populations during the course of the feedlot period are shown in Table 2 and Table 3. None of the main phyla (i.e., the ones with abundance greater than 0.1%) significantly changed (*p* ≥ 0.26) in the feces of the less efficient steers during the feedlot phase; however, in the more efficient steers, there was a trend (*p* = 0.096) for a decrease in abundance of *Proteobacteria* during this same period. Apart from that change, no other variations were observed (*p* ≥ 0.17) in the fecal microbiota of the steers at the phylum level.

At the genus level of classification (Table 2 and Table 3), abundance of *Turicibacter* was increased (*p* = 0.01) in the hindgut of the most efficient steers, while, in the less efficient steers, the presence of *Clostridium* (*p* = 0.01) and *Oscillospira* (*p* = 0.04) were increased during the feedlot period.

In the less efficient (high RFI) steers, the population of *Clostridiaceae* increased (*p* = 0.01) and *Veillonellaceae* tended to increase (*p* = 0.07; Table 2). However, the population of *Ruminococcaceae* decreased (*p* = 0.02) during the feedlot phase in the feces of this group of steers. In the most efficient steers (low RFI), abundance of the families *Peptostreptococcaceae* (*p* = 0.03) and *Turicibacteraceae* (*p* = 0.01) were increased during the feedlot period, and there was a trend for *Mogibacteriaceae* to be increased *(p* = 0.08). A summary of the main bacterial families (i.e., with relative abundance ≥ 0.3%) are presented in Figure 2 and Figure 3.

## 4. Discussion

### 4.1. Animal Performance

Steer growth rate (Average Daily Gain) and final carcass weights were similar, regardless of feed efficiency classification. However, the daily dry matter intake was significantly lower in the most efficient steers compared to their less efficient counterparts (10.89 kg/d vs. 13.02 kg/d, respectively). Since both groups had similar body weight gains, but one group required lower dry matter intake to achieve the same level of gain, that group ended up being classified as the low RFI group of steers. Steers with differing feed efficiencies, assessed as RFI, did not have differences in average daily gain and most of their carcass traits, including carcass weight [26]. Likewise, Hegarty et al. [26] found that steers with divergent RFI values had very different daily dry matter intake but maintained similar average daily gains. Therefore, the animal performance results observed in the present study are in harmony with previous research.

### 4.2. Changes in Microbial Diversity

The host animal can exert some control over its own GIT microbiome through several biological mechanisms, one of which is by secreting more substrates for the microorganisms to utilize [27]. Some bacteria that live close to the intestinal epithelial cells have adapted to that glycan-rich environment by producing more mucus-degrading enzymes [27]. Consequently, the host animals that produce more mucus in their intestines should have greater populations of bacteria specialized in mucus degradation. Moreover, since this kind of biological mechanism is inherent to each host animal, certain animals develop GIT microbiomes with unique characteristics, which may help them to be more or less efficient.

Microbial richness (measured as Chao 1 index) was increased by 18.13% in the feces of the most efficient steers during the 82-d feedlot period. In contrast, the Chao1 index increased only by 9.86%, the feces of the less efficient steers during the same period. Similarly, microbial diversity (Faith’s phylogenetic diversity) in the most efficient steers was increased by 10.7% during feedlot stage, whereas, in the less efficient steers, the increase amounted to only 0.3%. This greater increase in both microbial richness and diversity detected in the large intestine of the most efficient steers likely contributed to their superior conversion of feed into body weight gain. Digesta that reaches the large intestine in ruminants is comprised of feedstuffs from the previous compartments of the GIT tract that resisted ruminal microbial degradation and host intestinal digestion, so this digesta contains a higher percentage of fiber and other low-digestibility components than the original diet did [28]. Increased hindgut microbial richness and diversity in the efficient steers likely resulted in a greater number and variety of enzymes, equipping the microbiota with the ability to degrade fibrous digesta to a greater extent, thus extracting more energy from the digesta. Our present results are in line with those who found that microbial richness and diversity were greater in both the cecum and feces of beef cattle that had higher feed efficiency as measured by RFI [17].

### 4.3. Changes in Specific Microbial Taxa in the Hindgut

At the phylum level, the more efficient steers experienced a significant decrease in the abundance of *Proteobacteria* in their hindgut during the feedlot phase. More specifically, their population of *Proteobacteria* dropped to less than half of what it was at the beginning of the feedlot period (0.80% at d 0, and 0.36% at d 82). *Proteobacteria* are recognized as a phylum of bacteria that contains a wide variety of opportunistic pathogens, including the well-known *Escherichia*, *Campylobacter*, *Vibrio*, *Helicobacter*, and *Salmonella* [29], and some researchers have identified *Proteobacteria* as a “microbial signature” of disease [30]. *Proteobacteria* populations are frequently higher in individuals with endotoxemia and other metabolic disorders, and are often associated with endogenous alcohol production, which is a potential cause of liver damage [30]. In this context, the decrease in the population of *Proteobacteria* experienced by the most efficient steers in our study likely contributed to the better feed efficiency observed in this group of animals, by reducing opportunistic pathogenic burden on these cattle. This theory is supported because *Proteobacteria* in the less efficient group underwent a slight increase in abundance. Thus, the decrease in the population of *Proteobacteria* might have induced less inflammatory reactions in the most efficient steers compared to their counterparts, resulting in less energy being used in non-growth expenditures, thereby contributing to a better feed conversion of the high efficiency steers.

The abundance of two bacterial families were significantly increased in the feces of the most efficient steers during the feedlot phase: *P**eptostreptococcaceae* and *Turicibacteraceae*. In addition, *Mogibacteriaceae* populations tended to be higher in the feces of the most efficient steers during the feedlot phase. It has been demonstrated that some members of the *Peptostreptococcaceae* family can produce large amounts of ammonia from different nitrogen sources and are classified as hyper-ammonia-producing microorganisms [31]. Moreover, many bacteria that ferment structural carbohydrates prefer ammonia as their nitrogen source [32]. Therefore, a greater abundance of *Peptostreptococcaceae* may have contributed to a greater availability of ammonia in the hindgut of the most efficient steers, allowing more structural carbohydrate-fermenting bacteria to thrive.

The increase in abundance of *Mogibacteriaceae* in the most efficient steers is similar to the results reported by Fu et al. [33], who found that this family was positively correlated with plasma total cholesterol and increased body weight gain in mice [33]. In Wagyu beef cattle, Abbas et al. [34] reported increased *Mogibacteriaceae* populations in the rumen of cattle with higher marbling (intramuscular fat) scores. Myer et al. [12] found differences in the abundances of two members of the family *Mogibacteriaceae* in steers with different feed efficiencies, with the abundances being greater in the less efficient steers; however, while the present study analyzed fecal samples, Myer et al. [12] analyzed ruminal samples, which may explain the divergence between our study and theirs.

Cattle from the most efficient group in our study had their fecal population of *Turicibacteraceae* increased during their period in the feedlot. Tang et al. [35] reported a large variability in the population of *Turicibacteraceae* in the feces of dairy cows, whereas Wang et al. [36] reported that this family significantly affected growth performance in pigs. In mice, Henning et al. [37] observed that *Turicibacter* was positively correlated with body weight gain. In beef cattle, Myer et al. [16] found greater abundance of *Turicibacter* in the cecum of steers with greater feed efficiency (i.e., the ones that had less feed intake and greater average daily gain), indicating that the population of this family in the hindgut of cattle may be linked with feed efficiency. Our findings are similar to those of Myer et al. [16], because our most efficient steers had a significant increase in their fecal populations of *Turicibacteraceae* (and the genus *Turicibacter*) over the course of the feedlot period. Therefore, collectively, these results support the idea that the population of *Turicibacter* in the hindgut of cattle may be linked with feed efficiency.

*Oscillospira* is frequently detected by 16S rRNA gene surveys, however, this bacterial genus has not been previously cultured, so little is known about their metabolism, physiology, and ecological role [38]. Recent studies in humans have shown that *Oscillospira* is highly inheritable and is associated with leanness (i.e., lower body mass index). Additionally, *Oscillospira* species do not appear to degrade complex fiber, but rather rely on degradation products from microbial degradation of feeds, including intermediate fermentation products from primary fiber colonizing bacterial species [39]. Furthermore, Konikoff and Gophna [39] have hypothesized that *Oscillospira* was associated with leanness, and may even contribute to it, because they utilize the sugars liberated from the degradation of mucin produced by the host, which in turn forces the host to spend more energy to regenerate the degraded glycoproteins comprising the intestinal mucin [39]. We propose a potentially similar biological mechanism may have occurred in the current feedlot steers, which would explain the increase in *Oscillospira* in the less efficient, but not the most efficient steers.

## 5. Conclusions

In the present study, steers with divergent feed efficiency (assessed as residual feed intake–RFI) had marked differences regarding their feed intake while achieving the same level of body weight gain during the feedlot-finishing period. Investigation of their fecal microbiomes indicated that those differences in feed efficiency might be at least partially due to differences in their intestinal microbial populations, as several microbial taxa were found at different abundances between the two groups of steers. In addition, the most efficient steers underwent an increase in microbial richness and diversity during the feedlot period, whereas the less efficient steers did not have such changes. Such increases in microbial richness and diversity likely resulted in a greater amount (and array) of fiber-degrading enzymes in the hindgut of the most efficient steers, allowing them to extract more energy from the fibrous material that reached their hindgut, which ultimately increased their feed efficiency. Our findings may be crucial to beef producers, since feeding costs account for the largest input cost in beef operations. By learning more about which bacterial families are linked with better feed efficiency, researchers can begin to elucidate what makes an “ideal” gastrointestinal microbiota. However, due to the small sample size utilized in this study, results should be interpreted with caution, and further research should be carried out to confirm these findings and to investigate deeper taxonomic levels.

## Figures and Tables

**Figure 1 microorganisms-10-01128-f001:**
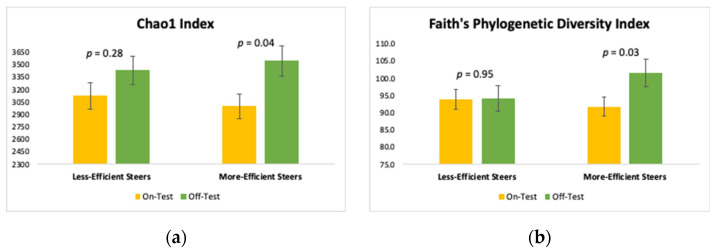
Chao1 index (microbial richness estimator) (**a**) and Faith’s Phylogenetic Diversity Index (microbial diversity estimator) (**b**) calculated for the less efficient (High RFI) and more efficient (Low RFI) steers at the beginning (d 0) and end (d 82) of the feedlot phase.

**Figure 2 microorganisms-10-01128-f002:**
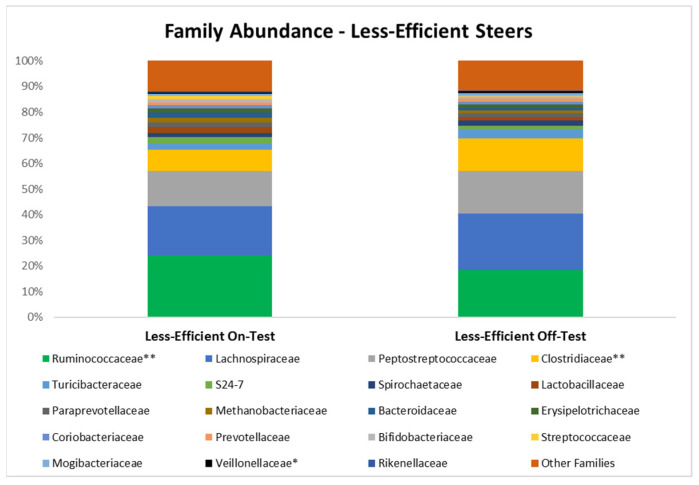
Relative abundance of the main families detected in the feces of the less efficient (High RFI) steers at the beginning (on-test; d 0) and end (off-test; d 82) of the feedlot phase. * Indicates a trend (0.05 < *p* ≤ 0.10). ** Indicates a difference (*p* ≤ 0.05) in abundance of the family.

**Figure 3 microorganisms-10-01128-f003:**
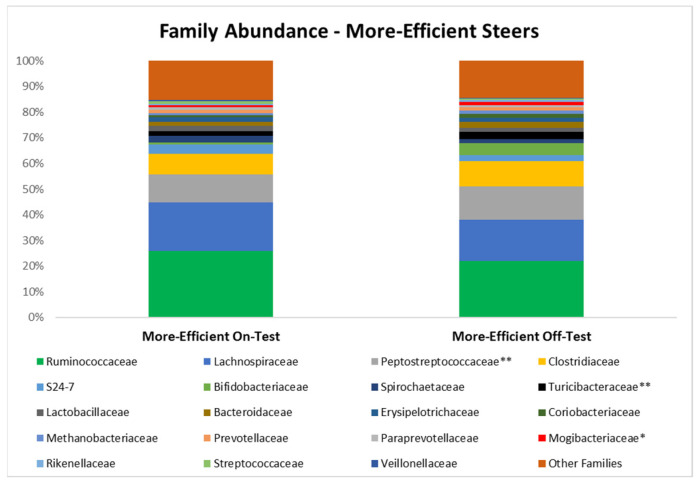
Relative abundance of the main families detected in the feces of the more efficient (Low RFI) steers at the beginning (on-test; d 0) and end (off-test; d 82) of the feedlot phase. * Indicates a trend (0.05 < *p* ≤ 0.10). ** Indicates a difference (*p* ≤ 0.05) in abundance of the family.

**Table 1 microorganisms-10-01128-t001:** Animal performance data observed during the feedlot-finishing period for steers differing in feed efficiency (assessed as RFI: Residual Feed Intake).

Item	Steer Classification		*p*-Value ^1^
	Inefficient (High RFI)	Efficient (Low RFI)	
Average daily gain, kg/day	1.05	1.02	0.82
Dry matter intake, kg/day	13.02	10.89	0.03
Residual Feed Intake, kg/day	0.76	−1.09	0.003
Hot carcass weight, kg	367.9	378.6	0.62

^1^*p*-value for the contrast between high and low-RFI steers.

**Table 2 microorganisms-10-01128-t002:** Summary of all bacterial taxa found to be significantly changed (or with tendency to change) in the less efficient (High RFI) steers from beginning to end of the feedlot period.

Taxa	**Abundance during Feedlot**		***p*-Value**
	Beginning (%)	End (%)	
Family *Ruminococcaceae*	24.01	18.35	0.02
Family *Clostridiaceae*	8.49	12.76	0.01
Family *Veillonellaceae*	0.34	0.63	0.07
Genus *Clostridium*	0.86	1.12	0.01
Genus *Oscillospira*	0.53	0.83	0.04

**Table 3 microorganisms-10-01128-t003:** Summary of all bacterial taxa found to be significantly changed (or with tendency to change) in the more efficient (Low RFI) steers from beginning to end of the feedlot period.

Taxa	Abundance during Feedlot		*p*-Value
	Beginning (%)	End (%)	
Phylum *Proteobacteria*	0.80	0.36	0.096
Family *Peptostreptococcaceae*	10.75	12.94	0.03
Family *Turicibacteraceae*	1.59	2.81	0.01
Family *Mogibacteriaceae*	0.63	1.17	0.08
Genus *Turicibacter*	1.59	2.81	0.01

## Data Availability

The data used to support the findings of this study are available from the corresponding author upon request.

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
