# Peer review of "Fecal Microbiome Differences in Angus Steers with Differing Feed Efficiencies during the Feedlot-Finishing Phase"

_microorganisms, 2022, doi:10.3390/microorganisms10061128_

Round 1

Reviewer 1 Report

The study is very interesting and have obvious effort but only I have the following inquiries:

1- Is there any data on health state of the less efficient animals specially in the paper you stated that the opportunistic pathogens is increased in this group?

2- Why you did not collect fecal samples in midpoint time (41 day) specially you already had measures for the body weight at this time point? May be it will be explain why you did not do this or even explain it as a limitation.

3- I hope you explain for the reader the relation of the results of your study to the steer breeders and the economical importance of your study

Author Response

Comments and Suggestions for Authors

The study is very interesting and have obvious effort but only I have the following inquiries:

1- Is there any data on health state of the less efficient animals specially in the paper you stated that the opportunistic pathogens is increased in this group?

Unfortunately, there was not health data collected on the steers throughout the feedlot trial. However, the steers were observed daily by the farm workers for any clinical signs of disease, and if any were observed the animals were removed from the study and treated. Additionally, we can infer from there average daily gain being similar to the efficient steers and their feed intake being increased that they were not undergoing any signs of disease (e.g., going off feed) that would’ve resulted in a decrease in animal performance.

2- Why you did not collect fecal samples in midpoint time (41 day) specially you already had measures for the body weight at this time point? May be it will be explain why you did not do this or even explain it as a limitation.

During the feedlot trial, the steers were at a feedlot in North Carolina, so we had to travel to collect the samples at the beginning and end of the feedlot trial. The farm workers collected the weights, so we were not there to collect fecal samples. Only having two time points may be identified as a limitation that needs to be explored in the future; still, the results from this study highlight the importance of the fecal microbiota in terms of host feed efficiency. In future studies of the effect of feed efficiency during the feedlot-finishing trial though, more data points may be collected (e.g., midpoint) to get a better understanding of how the microbial populations are changing.

3- I hope you explain for the reader the relation of the results of your study to the steer breeders and the economical importance of your study

In order to stress the importance of these findings in terms of breeding and economic impact, some sentences were included in the conclusion, and they read: “Our findings may be crucial to beef producers since feeding costs account for the largest input cost in beef operations. By learning more about which bacterial families are linked with better feed efficiency, researchers can begin to elucidate what makes an “ideal” gastrointestinal microbiota. However, due to the small sample size utilized in this study, results should be interpreted with caution; and further research should be carried out to confirm these findings, and to investigate deeper taxonomic levels.”

Reviewer 2 Report

The strong point of this manuscript is the subject studied. The research topic is very interesting and current. There are few data in the literature on the subject. However, in my opinion, there are few data for the conclusion presented. The number of animals for microbiome analysis was low; it was not possible to identify the main genus with the methodology used; the design is confusing. In my opinion, the manuscript has no merit to be published in a journal with the impact of Microorganisms, unless the authors improve significantly and add more information. .

some suggestions:
define the abbreviations in the abstract
L 55: populations or diversity?
L84-87: how and where was the collection performed? in the rectum?
L123: Please further explain the design used for the analyzes related to the performance and microbiome study. What design is used? Which mathematical model? Was time a factor in the study?
Why wasn't it possible to identify at the genus level? or the data was not placed?
L 22-23: Why does diversity increase fiber digestion? Are these microorganisms degrading the cell wall?
L53-56: where are these correlation analysis?
L76-77: Are these data shown?
I could not access the supplemental data

Author Response

Comments and Suggestions for Authors

The strong point of this manuscript is the subject studied. The research topic is very interesting and current. There are few data in the literature on the subject. However, in my opinion, there are few data for the conclusion presented. The number of animals for microbiome analysis was low; it was not possible to identify the main genus with the methodology used; the design is confusing. In my opinion, the manuscript has no merit to be published in a journal with the impact of Microorganisms, unless the authors improve significantly and add more information.

Thank you for your comments. We agree that the study has some limitations, including a small sample size; however, the findings are interesting: greater intestinal richness and diversity observed in the hindgut of the steers with best feed efficiency may be one of the reasons why they had better feed efficiency. We believe this finding alone is very important, and worth of being published, despite the limitations of the study, since the cost of feeding animals typically corresponds to over 70% of the total cost. Therefore, better feed efficiency is always a desirable trait, and having biological findings that support better feed efficiency makes the study relevant. However, because we agree with the reviewer that the study has limitations, we included a statement in our conclusion, which reads: Despite the interesting results, due to the small sample size utilized in this study, results should be interpreted with caution; and further research should be carried out to confirm these findings.

some suggestions:

define the abbreviations in the abstract

The abbreviations used in the abstract have been defined at first use.

L 55: populations or diversity?

In order to clarify what the authors meant in this sentence the addition of “…populations as a whole and individual families…” was added to the text to show that it is both the overall composition (diversity) and also specific families themselves impact efficiency.

L84-87: how and where was the collection performed? in the rectum?

In order to clarify the methods used for fecal collection, the M&M have been adjusted and now read, “Fecal samples were obtained aseptically via fecal grab from the rectum using separate palpation sleeves from all steers…” in order to show that the samples were collected from the rectum, aseptically.

L123: Please further explain the design used for the analyzes related to the performance and microbiome study. What design is used? Which mathematical model? Was time a factor in the study?

As requested, we added more information to that section. Animal performance data such as average daily body weight gain, dry matter intake, etc., were analyzed by one way ANOVA. The microbial abundances and diversity indexes were analyzed by paired t-tests, because the measures were done on the same animal at the beginning and end of the feedlot period (d0 and d82); and the results are shown for each efficiency group. So, yes, time was a factor.

Why wasn't it possible to identify at the genus level? or the data was not placed?

We did not show data at the genus level because a large percentage of our samples were not classified at this taxonomic level, resulting in many “undefined” genera. On the other hand, because most samples were classified at the family level, we decided to use that level for this manuscript.

L 22-23: Why does diversity increase fiber digestion? Are these microorganisms degrading the cell wall?

Correct. They break down cell wall.

L53-56: where are these correlation analysis?

The word “correlation” in that context simply means “association”. To avoid confusion, we now used the word “associations” in this new version of the manuscript.

L76-77: Are these data shown?

I could not access the supplemental data

A summary of animal performance is shown in Table 1: Average daily body weight gain during the feedlot period, average dry matter intake, etc. However, we included one additional table (supplemental table S1) that contains several of the animal performance data measured on those animals, for each individual animal.

Round 2

Reviewer 2 Report

Considering the importance of the topic, the changes in the manuscript and also the response to the comments review, I consider that the manuscript should be accepted for publication